# Comparative Osteogenesis and Degradation Behavior of Magnesium Implant in Epiphysis and Diaphysis of the Long Bone in the Rat Model

**DOI:** 10.3390/ma15165630

**Published:** 2022-08-16

**Authors:** Nhat Tien Tran, Yu-Kyoung Kim, Seo-Young Kim, Min-Ho Lee, Kwang-Bok Lee

**Affiliations:** 1Department of Orthopedic Surgery, Research Institute of Clinical Medicine of Jeonbuk National University, Biomedical Research Institute of Jeonbuk National University Hospital, Jeonbuk National University Medical School, Jeonju 54896, Korea; 2Department of Surgery, Hue University of Medicine and Pharmacy, Hue University, Hue 530000, Vietnam; 3Department of Dental Biomaterials, Institute of Biodegradable Materials and Oral Bioscience, School of Dentistry, Jeonbuk National University, Jeonju 54896, Korea

**Keywords:** magnesium implant, cortical bone, cancellous bone, epiphysis, diaphysis, gas formation, bone regeneration

## Abstract

Magnesium (Mg), as a biodegradable material, is a promising candidate for orthopedic surgery. Long-bone fractures usually occur in cancellous-bone-rich epiphysis at each end or the cortical-rich diaphysis in the center, with different bone healing processes. Little is known about the differences in results between the two regions when applying Mg implants. Therefore, this study aimed to compare the biodegradation and osteogenesis of Mg implants in a rat model’s epiphysis and diaphysis of the long bone. Twelve male Sprague Dawley rats underwent Mg rod implantation in the distal femoral epiphyses and tibial diaphyses. Every three weeks for up to twelve weeks, degradation behavior, gas evolution, and new bone formation were measured by micro CT. Histomorphology was analyzed by Hematoxylin and Eosin, Villanueva bone staining, and TRAP staining for osteoclastogenesis evaluations. Micro-CT analysis showed statistically significant higher new bone formation in the epiphysis group than in the diaphysis group, which correlated with a lower gas volume. Histological analysis showed higher osseointegration of Mg implants in the epiphyseal region than in the diaphyseal region. The magnesium implant’s osteoclastogenesis-inhibiting properties were shown in the surrounding areas in both the cortical bone of the diaphysis and the cancellous bone of the epiphysis. Our findings show the differences in the magnesium implant’s osteogenesis and biodegradation in the epiphysis and the diaphysis. These dissimilarities indicate a better response of the epiphyseal region to the Mg implants, a promising biomaterial for orthopedic surgery applications.

## 1. Introduction

Biodegradable metals are considered promising biomaterials for medical applications [1]. Among the metals, magnesium and its alloys are most popular owing to their suitable biocompatibility, biodegradation, and biomechanical properties. As the fourth most abundant cation in the human body, magnesium is mainly stored in the bone matrix and is essential for a wide range of metabolic processes, stimulating bone growth, and facilitating tissue healing [2]. Magnesium alloy implants are degraded under the physiological environment, therefore eliminating the need for secondary surgeries for implant removal. Furthermore, the corrosion product of the degrading process is Mg^2+^, which does not cause unforeseen complications because extra magnesium ion is permissible and can be eliminated along with urine and feces without leaving residual cellular toxicity in the human body [3]. Magnesium has a higher strength relative to natural bone, with the elastic modulus value close to that of the cortical bone, reducing stress shielding at the interface of the implant to the bone during load transfer. Therefore, magnesium is a potential biocompatible, biodegradable, lightweight, and load-bearing orthopedic implant [4,5,6,7].

Many in vitro and in vivo studies have shown that novel magnesium materials with optimized mechanical and biological properties are potential candidates for clinical application, especially in bone healing and remodeling [8]. Biodegradable magnesium-based implants have been applied in bone fractures, including the distal humerus, ulnar, scaphoid, patella, malleolar fractures, intra-articular fractures, and osteotomies around the foot and ankle [9]. For magnesium-based implants, their most significant disadvantage is the high corrosion rate with resultant rapid loss of mechanical integrity, hydrogen gas evolution, and alkalization at the degradative time. Various strategies, including alloying and surface coating, have been employed to overcome this limitation. In general, magnesium implants are indicated in osteosynthesis for small to medium-sized bones or epiphysis of the long bone. Meanwhile, the lack of Mg implants in the diaphysis of long weight-bearing bones, such as femur and tibia shaft fractures, is attributed to early complications in clinical applications [8,10]. So, there are still some challenges in applying biodegradable magnesium-based metal in orthopedic surgery, especially for weight-bearing skeletal sites [2].

The mature human skeleton has two distinct types of bone based on their porosities: cortical and cancellous bones. Found predominantly in the diaphysis of long bones, cortical bones comprise 80% of the skeleton and have a highly dense and slow turnover structure with high resistance to bending and torsion. Cancellous bones, also known as trabecular bones or spongy bones, make up 20% of the skeletal mass, and are mainly found at the epiphysis of long bones, spinal vertebrae, pelvis, and other large flat bones. Compared to cortical bones, cancellous bones are more elastic, less dense, and have higher metabolic activities such as bone regeneration and mineral exchange [11]. Both cortical and cancellous bones are essential to bone strength and bone remodeling with a complicated relationship [12]. The previous studies demonstrated that magnesium has osteogenic effects on the cortical and cancellous bones in long bones [13]. However, cancellous bone and cortical bone have different histology and remodeling processes that result in an acute form of inflammation in the cancellous bone healing and a chronic type of inflammation in the cortical bone healing. Previous experimental models showed that cancellous bone healing has a cellular pattern of classical acute inflammation. On the contrary, cortical healing has a more chronic inflammatory process [14]. In humans, fractures are prominent in the epiphysis regions of long bones, such as the proximal or distal regions of the humerus or femur, distal radius, and proximal tibia. However, most studies on biodegradable magnesium implants have focused on the cortical-rich diaphysis in the center [2]. Several studies also inserted magnesium-based biodegradable implant into the epiphysis of the long bone, which showed excellent biocompatibility and homogeneous degradation characteristics with negligible production of hydrogen gas [15,16].

While magnesium implants have favorable biodegradation, biocompatibility, and osteoconduction to bone regeneration, no in vivo comparison studies have analyzed these characteristics in different regions of the long bone. We hypothesized that there are differences in gas formation and bone remodeling around the magnesium implant between the cortical bone in the diaphysis and cancellous bone in the epiphysis. Herein, we used adolescent rats to conduct a descriptive study to compare the biodegradation and osteogenesis of magnesium implants in the epiphysis and diaphysis of the long bone.

## 2. Materials and Methods

### 2.1. Experimental Design

The current study was carried out according to the Declaration of Helsinki and was approved by the Institutional Animal Care and Use Committee of the Chonbuk National University Laboratory Animal Center (CBNU 2020-096). 

The chemical components of the Mg rod (Mg 99.93%, Sincere East Foreign Trade Corp., China) were Mg 99.93, Si 0.0228, Mn 0.0128, Al 0.0032, Fe 0.0017, Cu 0.0005, and Ni 0.0003 (wt%). The magnesium rod was cut and sequentially ground using SiC sandpaper with grits ranging from #600 to #2000. All the samples were cleaned with distilled water and 100% ethanol, dried in a stream of dry air and sterilized with ethylene oxide (ETO) for at least 8 h at 40 °C and 1.7 bar of atmospheric pressure. Forty-eight cylinder magnesium rods: 1.6 mm × 4 mm (Figure 1a,b) were used for implantation in twelve 8-week-old male Sprague Dawley rats (weight: 260–280 g). The diaphyseal tibial shaft was used to evaluate the magnesium implant’s osteogenesis and biodegradation behavior in the cortical bone, while the characteristics of cancellous bone were studied at the distal femoral epiphysis.

### 2.2. Surgical Procedures

Four magnesium rods were implanted on both sides of each rat: two from the lateral to medial positions at tibial diaphysis and two from the distal to proximal regions of the distal femoral epiphyses (Figure 1c,d). An intramuscular dose of 50 mg/kg of Tiletamine plus Zolazepam (Zoletil 50, Virbac Laboratories, Carros, France) and 15 mg/kg of Xylazine Hydrochloride (Rompun, Leverkusen, Germany) were used to anesthetize the rat. The surgical areas were shaved and disinfected with iodine scrubs, and 1.5 cm incisions were made on the para-patella from the lower thigh to the middle-third leg. The incisions surgically exposed the distal femoral condyle and the lateral tibial shaft through the skin and fascia. The periosteum was then carefully separated from the bone after a full-layer incision with a #11-blade and a periosteal elevator.

After flap reflection, a contra-angle handpiece (X-smart Endodontic Motor, Dentsply Maillefer, Ballaigues, Switzerland) was used to carve a hole in the bone and prepare an implant bed in the lateral region of the tibia diaphysis with a 1.6 mm pilot roundheaded bur (H1.316-018, Komet, Lemgo, Germany). A similar procedure was followed for the epiphyseal region of the distal femur. Then, the implants were inserted into their designated positions (Figure 1). The operation was conducted under copious saline irrigation and performed at a rotary speed of ≤350 rpm. 

The soft tissue was adjusted and layered closed using an absorbable suture, and then the skin was closed with a non-absorbable nylon suture. Amikacin (1.5 mg/kg) was administered intramuscularly 0, 24, and 48 h after the operation. During the postoperative period, the animals could move around in their cages without assistance. Skin sutures were removed on day ten postoperation. Triweekly postimplantation, three rats were sacrificed for micro-CT and histological assessment by euthanasia using an overdose of thiopental sodium (ChoongWae Pharma Corporation, Seoul, Korea).

### 2.3. Micro-Computed Tomography Analysis

A trans-axial cutting 0.5 cm above and below the implantation point was used to harvest the bone specimens, together with the magnesium rods. Block biopsy specimens were quantitatively analyzed using A SKYSCAN 1076 Micro-CT unit (Kontich, Belgium) at 100 kV and 100 μA, with 240 ms of exposure time. The specimens were immersed in a 10% neutral buffer formalin solution during imaging. The phantom in Hounsfield Units (HU) using CTAn software version 1.16.4.1 (Skyscan) was used to estimate the bone and gas formation volumes. The micro-CT scan image was reconstructed by DataViewer software version 1.5.2.4 and then analyzed with CTAn software. A worldwide thresholding calculation was connected at a consistent edge for all examples. The region of interest (ROI), commonly defined as adjacent to the implant and endo-cortical surface for trabecular bone, specifies the area to be evaluated from the reconstructed images. The ROI for analyzing material degradation was a cylinder of the same size and position as the rod. A new larger ROI in the same shape and position was used to monitor new bone development and assess the stimulatory effects of magnesium alloys on bone growth (Figure 2). Bone volume fraction (BV/TV) is the volume of mineralized bone within a specific tissue volume of interest. Gas volume was measured in the same manner as the bone volume, but the bone mineral density (BMD) calibration value was different.

### 2.4. Histological Analysis 

After micro-CT scanning, each group’s blocks underwent a series of fixation, staining, and embedding procedures to undertake the histological examination. The blocks were fixed in fresh 10% formalin for two days. The staining method then determined the subsequent ongoing processes.

Villanueva bone staining: the fixed femurs and tibia blocks were immersed in Villanueva solution (Polysciences, Inc., Eppelheim, Germany) for three days. Next, the blocks were dehydrated with gradient ethanol (80, 90, 95, and 100%) and 100% acetone. For embedding in resin, the blocks were pre-permeated with methylmethacrylate (MMA, Yaruki Pure Chemicals Co., Ltd., Kyoto, Japan) under vacuum for 2 h and then infiltrated with the polymerization mixture (PMMA) at 35 °C for 3 days followed by 60 °C for 1 day. Prepared resin blocks were cut into 0.5 mm slices through the implant midline using a low-speed saw (EXAKT 300 CP, EXAKT Technologies Inc., Norderstedt, Germany). For the histological analysis, slices were ground to a 70 µm thickness using a micro-grinding system (EXAKT 400 CS, EXAKT Technologies Inc., Norderstedt, Germany).

H&E staining, TRAP staining: the specimen blocks were decalcified in 10% EDTA (Ethylenediaminetetraacetic acid) for 28 days. Subsequently, the magnesium implant was carefully removed, and the bone surrounding the implanted area was sectioned and embedded in paraffin following the standard procedure. Serial sections of paraffin-embedded decalcified bone tissue (5 µm thick) were collected from the center line of the defect. Using hematoxylin and eosin (H & E) staining (Mayer’s Hematoxylin and Eosin Y Solution, Sigma-Aldrich, St. Louis, MO, USA), morphological observation and morphometry of new bone were performed to confirm pathological reactions around the materials. Tartrate-resistant acid phosphatase (TRAP) staining (Sigma-Aldrich#387A) was performed to evaluate osteoclastogenesis around the implanted material. Images were acquired with an optical microscope (Leica DM750, Wetzlar, Germany). 

### 2.5. Statistical Analysis

SPSS software version 25.0 (IBM, Armonk, NY, USA) was used to conduct a statistical analysis of the data. Comparisons between two groups were evaluated using an unpaired two-tailed Student’s *t*-test, and differences with a *p*-value < 0.05 were considered statistically significant.

## 3. Results 

All the 12 rats’ wounds healed smoothly and without infections or local inflammation.

### 3.1. Micro-CT Evaluation

The volume and surface changes of the magnesium rods and the amount of gas formation were continuously evaluated by micro-CT analysis throughout the study. Figure 3 presents the axial and sagittal plane micro-computed tomography showing the postoperative changes in each group for up to 12 weeks, demonstrating the gas and new bone formation around the implant. Early gas formation around the rods was evident in all the animals by the third week postoperation, and the gas expansion areas were larger after six weeks. A substantial gas bubble appeared in the intramedullary cavity around the magnesium rods, visible in the micro-CT sections of Figure 3. Likewise, the gas volume was significantly higher in the diaphysis group compared to the epiphysis group (*p* < 0.001) 3, 6, and 9 weeks postoperatively. In addition, the gas bubbles were absorbed expeditiously, as the gas volume around the implants decreased from 6 to 12 weeks in each group and showed no significant difference at the 12-week time point (Figure 4b). The magnesium volume decreases with the initial corrosion time; it gradually decreases as it is absorbed into the body (Figure 4a). However, the degradation rate of magnesium implants in the diaphysis group was significantly higher than in the epiphysis group (*p* < 0.05). The micro-CT analysis demonstrated a significantly higher new bone volume formation around the magnesium implant in the epiphysis group compared to the diaphysis group (*p* < 0.001). From 3 to 12 weeks, the epiphysis and diaphysis group’s bone volume fraction (BV/TV) increased from 71.6% to 91.3% and 45.3% to 73.3%, respectively (Figure 4c).

### 3.2. Osseointegration Evaluation by Villanueva Bone Staining

Histological analysis by Villanueva bone staining showed a new bone−implant interface in all groups, and red arrows indicated new bone regeneration. Within 12 weeks, new bone growth surrounding the rod was enhanced, with tight contact with the implant at the cortical, cancellous, and medullary cavity sites. At 3 and 6 weeks postoperation, the epiphysis group showed significantly greater new bone formation than the diaphysis group. In addition, the osseointegration between the implant and new bone in the epiphysis group was more robust than in the diaphysis group. With yellow arrows indicating the gap between new bone layers and implant surface, smaller gaps were observed between the implant and newly formed bone in the epiphysis group than in the diaphysis group. The epiphysis group had smaller gaps between the implant and newly created bone than the diaphysis group, as indicated by yellow arrows in Figure 5. In both groups at week 9 and week 12 postoperatively, they demonstrated more regenerative bone around implants and better osteointegration between implants and host bone. However, the epiphysis group showed better osteogenesis and full integration between the implant and new bone than the diaphysis group (Figure 5).

### 3.3. Histologic Analysis for the Surrounding Bone Tissue by H & E Staining

Hematoxylin and Eosin staining showed tissue morphology around the implant (Figure 6). After three weeks, the epiphysis group exhibited prominent osteogenesis, with osteoblasts and osteocytes covering the surface of new bone. Conversely, less bone formation appeared around the magnesium rod in the diaphysis group. In the diaphysis group, gas bubbles generated from magnesium dissolution could be observed around the implant. After six weeks, the newly formed bone was denser in both groups, the interface between old and new bone became indistinct in the epiphysis, and the gas pocket was smaller in the diaphysis group. At 12 weeks postoperation, increased new bone formation was observed around implants in all the groups. However, the epiphysis group showed more robust osteogenesis when compared with the diaphysis group. At high magnification, no apparent inflammatory cells were observed in the cortical and cancellous bone surrounding the magnesium implants at 3-, 6-, and 12-week time points. In general, the magnesium implant showed good biocompatibility without inducing an inflammatory response in both regions of the long bones.

### 3.4. Osteoclastogenesis Analysis of the Bone Tissue around the Mg Implant

The effect of magnesium implants on the expression of osteoclast-specific markers (TRAP) in the surrounding bone was investigated. TRAP is linked to osteoclast migration to bone resorption sites and is thought to be involved in osteoclast development, activation, and proliferation. The effects of magnesium implants on mature osteoclast resorption activity were investigated, and the results are shown in Figure 7. Both cancellous bone in the epiphysis and cortical bone in the diaphysis contained the TRAP+ cells in the control areas. In particular, the former showed a higher number of TRAP+ cells than the latter. However, the presence of the magnesium rod resulted in a remarkable reduction in osteoclast resorption activity surrounding the implant in both epiphysis and diaphysis. At three weeks post magnesium implantation, the TRAP+ cells can be seen at the cancellous bone around the magnesium rod, but from six weeks after surgery, no TRAP+ cells were observed. This proved that the magnesium corrosion granules inhibit mature osteoclast cells’ activity. This has also been seen earlier at the cortical bone, as TRAP staining was negative since the three-week time point postoperatively.

## 4. Discussion

Magnesium-based implants have been shown as promising candidates for osteosynthetic use in orthopedics. Numerous studies have evaluated the degradation performance of the implant materials and their optimization. However, there are no studies comparing the different areas of a long bone, the impact of degradation products, gas production, and degradation rates on remodeling potential. Here, we compared the osteoconduction and biodegradable behavior of magnesium implants between epiphysis and diaphysis in rats and demonstrated differences in the biodegradation, osteogenesis, and osteoclastogenesis of Mg implants in the epiphysis and the diaphysis. The magnesium rod implanted into the rat could promote callus formation and osteogenesis in both epiphysis and diaphysis. However, the former shows higher osteogenesis with more new bone formation around the implant. On the other hand, the gas evolution induced is lower than the latter. The osteoclastogenesis-inhibiting properties of magnesium implants were shown in the adjacent surrounding areas in both cortical bone of the diaphysis and cancellous bone of the epiphysis. In particular, the osteoclast activity was perhaps suppressed faster in the cortical bone than in cancellous bone.

The complications of hydrogen gas produced from magnesium implants remain controversial. Some preclinical studies reported gas formation during degradation without specific complications [17,18,19]. However, other in vivo studies indicated that the rapid and persistent hydrogen gas cavity formation led to prolonged discomfort and affected blood cell formation, which decreased the survival rate of rats [20]. Hydrogen gas formation’s side effects on bone healing were mentioned in a few preclinical and clinical studies. The space-consuming gas pockets would raise the inner mechanical pressure and inhibit the initial bone healing process. The integration between the implant surface and newly formed osteocyte was also potentially restricted by the bubbles. Several studies describe this gas formation through magnesium biodegradation, which develops in the first week and decreases over time [19,21]. Meanwhile, Dang et al. demonstrated that the gas bubble remained in the rabbit models 12 weeks after magnesium screw implantation [22]. In our study, the micro-CT scans revealed that the gas evolution reached the peak value at six weeks and then went down to a tiny amount at 12 weeks. These results could be explained as the gas bubbles were absorbed into the surrounding tissue and replaced by the new bone of the osteogenesis process [23]. In particular, good healing results were achieved with the magnesium rod in the epiphysis with a lower degradation rate and reduced gas formation. On the other hand, the magnesium implant rapidly degraded within the diaphysis, and there was more gas evolution. This may be well-explained by the medullary cavity’s higher blood flow and water content in the diaphysis than the cancellous bone matrix in the epiphysis [24]. Three different systems supply a typical long bone, such as the femur or tibia: (1) a central nutrient artery, (2) epiphyseal arteries, and (3) periosteal arteries. The nutrient artery primarily nourishes the diaphysis, which passes through the cortex into the medullary cavity and sends upward and downward longitudinal bone marrow branches. It maintains high blood pressure to reach distant areas, usually by terminating numerous tiny vessels in the metaphysis and endosteum [25]. Moreover, the periosteal arteries penetrate and distribute blood to the outer third of the cortical bone of the diaphysis. In contrast, epiphyseal arteries supply the epiphysis region. This separate blood circulation enters the bone from a network of periarticular vascular plexus near the ends of long bones, which is smaller than the nutrient artery. Drainage veins of epiphyseal blood are also relatively smaller than those in the medullary region [26]. 

Magnesium implants may leave debris during degradation leading to local inflammatory responses and foreign body reactions [27]. In our study, no acute complications, such as wound infection, skin necrosis, or inflammatory reactions, were observed for up to 2 weeks postoperatively. The surgical position of the rats was followed until they were sacrificed and showed no evidence of local infections or inflammations observed even during harvesting samples. The histological analysis also showed no chronic inflammatory reaction (Figure 6). This evidence shows the biocompatibility of magnesium implants in both epiphysis and diaphysis of the long bone. Our results are similar to other animal experiments with the implantation of Mg with a screw, without a significant inflammatory reaction with the surrounding bone [19,28,29]. 

Our study found that osteogenic cells and bone calluses form around the magnesium implant earlier and are more generous in the epiphysis than in the diaphysis, as evidenced by micro-CT and histology. The bone regeneration characteristics of the two regions strongly support that the magnesium implant shows osteoconductive and osteoinductive properties in epiphysis better than in diaphysis of long bone [30]. Because cancellous bone is prevalent in the epiphyseal region, it includes more mesenchymal stem cells with high osteogenic potential than the diaphyseal region. This suggests that a bioabsorbable magnesium implant may promote greater osteogenesis in the epiphyseal area than in the cortical bone-rich regions, such as the diaphysis. On the periosteum side of the diaphysis, the typical bone repair process is separated into four stages: inflammation, soft callus development, hard callus formation, and remodeling [31,32]. On the other hand, the first stage of bone healing in the epiphysis is marked by a bleeding event, and inflammation is lower than in the diaphysis. The second and third stages of epiphyseal bone healing show mesenchymal stem cell activation and differentiation into osteoblasts and woven bone creation. The woven to lamellar bone transition defines the fourth stage, followed by ongoing bone remodeling in the final stage [33]. Research in mouse models has also indicated the acute type of inflammation in cancellous bone healing is more chronic in cortical healing, proving cancellous bone healing is faster than diaphyseal fractures [14]. 

Bone regeneration relies on the balance between osteogenesis and osteoclastogenesis, known as bone homeostasis, which is maintained by the osteoblast-mediated bone formation and bone resorption [34,35]. The excessive bone resorption by osteoclasts causes peri-implant osteolysis, which is the main reason for the poor long-term outcome of load-bearing orthopedic implants [36,37]. It is essential to develop metal materials with therapeutic osteoclast function to alleviate osteolysis and encourage osseointegration around the bone implants. Magnesium plays a crucial role in bone homeostasis. Low magnesium concentrations inhibit the osteoblast’s activity while promoting the osteoclast, resulting in inducing osteopenia [38,39,40]. However, the corrosion of magnesium implants results in the release of Mg^2+^ into the surrounding medium. Maradze et al. found that the concentration of magnesium corrosion products affects the in vitro response of osteoblast progenitors and cells in the osteoclastogenic lineage. High magnesium ions reduce the fusion of pre-osteoclast cells and the resorption activity of mature osteoclast cells [41]. The red marrow is the source for osteoclast precursors that reach the bone surface, wherever remodeling is needed via the circulation, not by direct migration. Cancellous bone has higher blood flow, surface-to-volume ratio, and turnover than cortical bone [12,42]. Thus, the osteoclast activities may differ between the epiphyseal and diaphyseal regions when applying magnesium implants. Because pre-osteoclasts first differentiate into TRAP-positive mononuclear cells during differentiation, TRAP staining is the gold standard for identifying osteoclasts [43]. In the present study, osteoclast activities were shown at both the epiphysis and diaphysis of the long bone, primarily located on the bone resorption surface of the cortical bone. For the first time, magnesium implants inhibited the osteoclastogenesis of the surrounding bone tissue in a rat model, which is only supported by previous in vitro research [41]. The osteoclast activity was suppressed three weeks post magnesium implantation in the diaphysis but later in the epiphysis. This might have been caused by the higher corrosion rate of magnesium rod in the diaphysis, which induced the higher Mg^2+^ concentration in the local media and stimulated the rapid inhibition of osteoclast activity.

There are some potential limitations of this study related to the interpretation of the data presented. Firstly, our study compared osteogenesis of diaphyseal and epiphyseal regions from two long bones: the tibial shaft and the distal femur. The distal femoral epiphysis and tibial diaphysis differ despite coming from the same lower leg, which was fully noted from the start. However, both regions being in the same leg could restrict the differences in the main artery supply to the limb, mobilization level, and other physiologic conditions. When designing the experiment with two regions on the same bone in a rat, they could affect each other through gas formation or any complication during bone regeneration: infection, inflammatory reaction, and implant migration. Thus, additional studies on the long bones of larger animals and human beings are needed to verify the supposed efficacy. Another limitation is that osteogenesis and degradation behavior were only evaluated for up to 12 weeks. The long-term degradation and biocompatibility results of the investigated magnesium implant will be addressed in further research. Moreover, further studies are warranted to evaluate magnesium device application in animal models by simulating the specific fracture situation of typical long-bone fractures.

## 5. Conclusions

The biodegradable magnesium implant exhibited good biocompatibility and osteogenesis in both epiphysis and diaphysis of the long bone. In comparison to the diaphyseal region, the epiphyseal region demonstrated better osseointegration, fresh bone creation, and homogeneous degradation characteristics with lower gas emissions. Therefore, this study shows that the biodegradable magnesium implant has excellent bone–implant integration and can thus be successfully used as an osteosynthesis device for fracture fixation in both epiphysis and diaphysis, with earlier bone healing in the epiphyseal regions at the ends of long bones.

## Figures and Tables

**Figure 1 materials-15-05630-f001:**
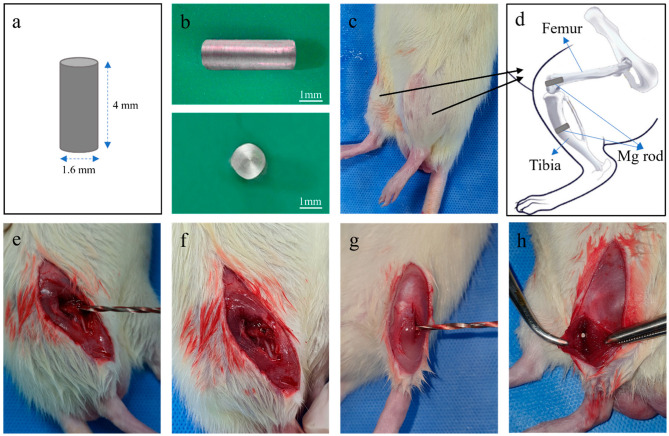
Cylinder rod type of the implant and position for in vivo test. (**a**,**b**) Schematic diagram and optical images of the magnesium implant used for the experiment, (**c**,**d**) sketch of 4-implant position at both rat’s lower limbs, (**e**,**f**) implant placement at distal femur, and (**g**,**h**) implant placement at tibia shaft.

**Figure 2 materials-15-05630-f002:**
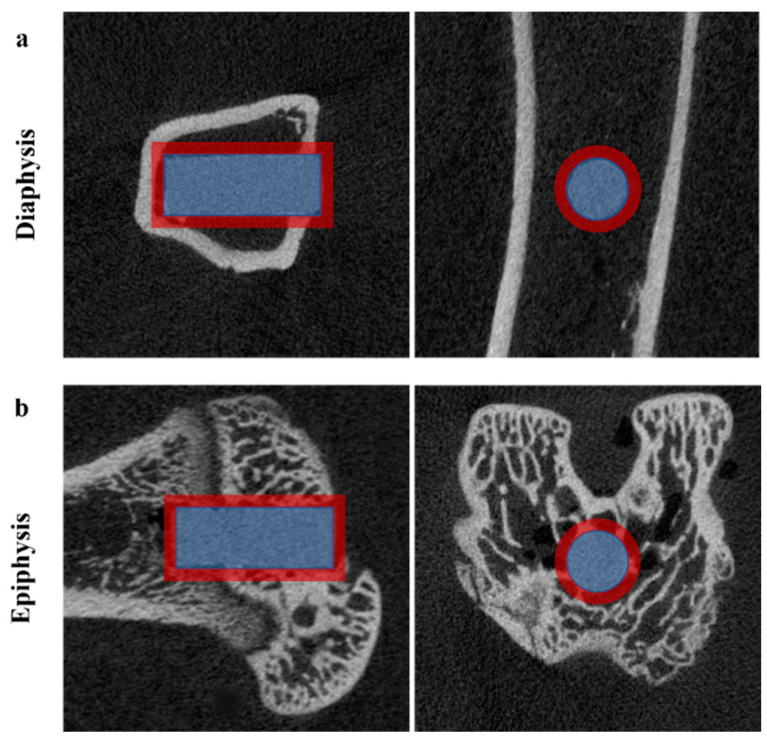
The region of interest (ROI) determines the volume of the Mg rod (blue cylinder) and new bone volume (red annular cylinder). The images from the micro-CT scan were analyzed by CTAn software in the axial and sagittal planes of the Mg implantations in (**a**) the tibial diaphysis and (**b**) distal femoral epiphysis.

**Figure 3 materials-15-05630-f003:**
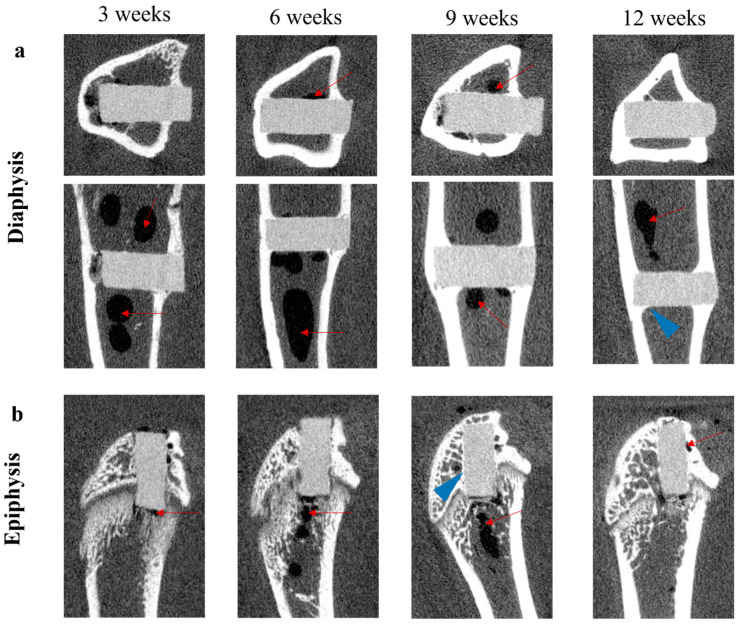
Micro-CT images of Mg rods in tibial diaphysis (**a**) and distal femoral epiphysis (**b**) after implantation in rats for 3, 6, 9, and 12 weeks. The red arrows indicate gas pocket formation, and the blue arrowheads indicate the new bone layers covering the implant surface.

**Figure 4 materials-15-05630-f004:**
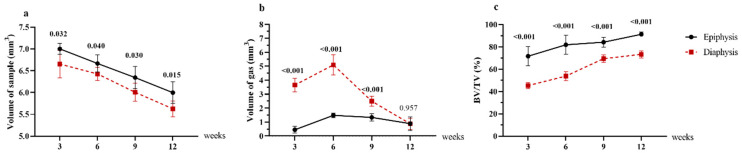
Quantitative analysis of micro-computed tomography. (**a**) The volume of the Mg rod, (**b**) gas pocket, (**c**), and bone volume fraction (BV/TV) around the implant, in the epiphysis (n = 6) and diaphysis (n = 6) groups at 3, 6, 9 and 12 weeks. Values are presented as the mean and standard deviation. Comparisons between two groups were evaluated using Student’s *t*-test, and *p* values are indicated at every time point. BV: new bone volume, TV: total tissue volume.

**Figure 5 materials-15-05630-f005:**
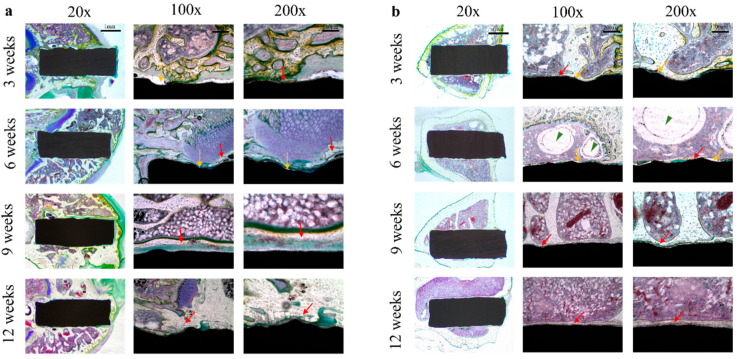
Histological images postoperatively (by Villanueva bone stain) at 3, 6, 9, and 12 weeks. (**a**) Distal femoral epiphysis, (**b**) tibial diaphysis. The red arrows indicate newly formed osteocytes, the yellow arrows indicate the gap between new bone layers and implant surface, and the gas pockets are marked as green arrowheads.

**Figure 6 materials-15-05630-f006:**
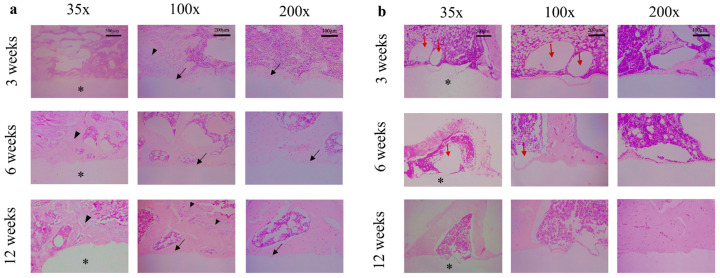
Hematoxylin and Eosin staining of the tissue surrounding the magnesium rod in (**a**) distal femoral epiphysis, and (**b**) tibial diaphysis at 3, 6, and 12 weeks. The black arrows indicate newly formed bone around the implant surface. The asterisk represents the position of the magnesium rod, and the gas pockets marked as red arrows. The black arrowhead shows the chondrocyte of the epiphyseal plate (growth plate).

**Figure 7 materials-15-05630-f007:**
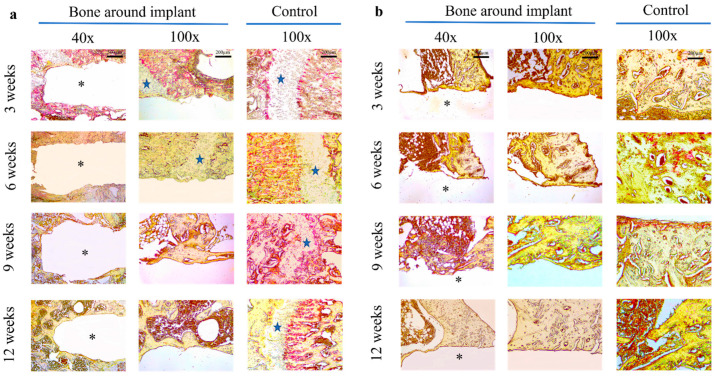
Mitigation of bone resorption surrounding magnesium implant in (**a**) distal femoral epiphysis and (**b**) tibial diaphysis at 3, 6, 9, and 12 weeks. TRAP (tartrate-resistant acidic phosphatase) staining of the bone surrounding the magnesium rod compared with the control area in the same bone. The asterisk indicates the position of the magnesium rod and the epiphyseal growth plate is marked as a blue star.

## Data Availability

Not applicable.

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
