# Peer review of "Comparative Osteogenesis and Degradation Behavior of Magnesium Implant in Epiphysis and Diaphysis of the Long Bone in the Rat Model"

_materials, 2022, doi:10.3390/ma15165630_

Round 1
Reviewer 1 Report
The reference list of the manuscript contains 43 titles, and is without inappropriate self-citations. The manuscript is clear, with a good rate of novelty and significance. The manuscript present scientific resound and the design appropriate to test the hypothesis. The methods and software are clear described, with sufficient details to permit another researcher to reproduce the results. All aspects regarding the figures/images are appropriate, and they are easy to interpret and understand. The presentation and the analyzed date are written in proper way. The presentation of the results are at high standard, with appropriate statistics. The results offer a development in the present knowledge, are significant, and are suitable interpreted.
Please check the unit "m" in section 2.4.:
...were cut, honed to 70 m thickness, and polished for optical...
Reviewer 2 Report
I thank the respected authors for this article entitled Comparative Osteogenesis and Degradation Behavior of Magnesium Implant in Epiphysis and Diaphysis of the Long Bone in the Rat Model.
I found few comments provided as stick-it notes on the uploaded pdf

Reviewer 3 Report
The paper deals with a relevant topic and reports a correctly planned and executed research. The manuscript quality is very good and consistent in all the chapters. The shortcomings are few and mostly represented by unclear passages of text. In particular:
Introduction, 11th line: the difference between stiffness and elastic modulus is subtle (they are often used as synonims) and may be better clarified.
Introduction, 7th-to-last line: “While” does not matter in this period and is perhaps a cut-and-paste remnant.
Materials and Methods, Histological analysis: this hasty section contrasts starkly with the very accurate description of the other techniques used. The authors should detail the make of the instrumentation used for ground sections, and explain how the implant was removed from the blocks used for microtome sectioning (H&E).
Results: the images are of very good quality but their great number implies they are quite small, which adversely affects their legibility. I understand that this can hardly be improved, but is nonetheless a pity.
Discussion, 21th line: it is not clear what the authors mean with the word “recommended”. This term usually means “encouraged”, “appreciated”. The subsequent paragraphs seem to convey the contrary.
All in all, the paper is very good and seems to require just a modest revision.
